# Mediational Occupational Risk Factors Pertaining to Work Ability According to Age, Gender and Professional Job Type

**DOI:** 10.3390/ijerph18030877

**Published:** 2021-01-20

**Authors:** Inmaculada Mateo-Rodríguez, Emily Caitlin Lily Knox, Coral Oliver-Hernández, Antonio Daponte-Codina

**Affiliations:** 1Andalusian School of Public Health (EASP), 18011 Granada, Spain; inmaculada.mateo.easp@juntadeandalucia.es; 2CIBER Epidemiology and Public Health (CIBERESP), 28209 Madrid, Spain; emily.knox.easp@juntadeandalucia.es; 3Faculty of Psychology, Complutense University of Madrid, 28040 Madrid, Spain; maoliver@ucm.es; 4Andalusian Observatory on Environment and Health (OSMAN), Andalusian School of Public Health, 18011 Granada, Spain

**Keywords:** work ability, healthcare workers, gender, age, occupational risk, mediation

## Abstract

The predictive value of work ability for several health and occupational outcomes is well known. Maintaining the ability to work of all employees has become an important topic in research although some evidence suggests that some groups of workers need greater attention than others. Healthcare workers ($\bar{x}$ = 54.46 ± 5.64 years) attending routine occupational health checkups completed their work ability, occupational risk and sociodemographic measures. An analysis examined whether work ability differed according to gender, age and professional category. Mediation of these relationships by occupational risk variables, such as work–family conflict, was examined. Females and older adults had worse work ability than their counterparts. Professional group was not independently associated. Gender-related differences were mediated by current and historic ergonomic risk, psychosocial risk and work–family conflict. Age-related differences were mediated by violence/discrimination at work. All job risk variables, apart from current ergonomic risk, mediated associations between professional category and work ability. The present study identified the importance of risk variables for the work ability of health workers according to gender, age and professional job type. Perceptions of work–family conflict and violence–discrimination seem particularly important and should be considered when targeting improvements in work ability.

## 1. Introduction

Work ability is defined as an individual’s perception of physical and mental demands encountered at work and his or her individual ability to cope with these work demands [1]. 

The ability to work should be considered an important characteristic of human health and well-being [2]. A number of studies show that low levels of work ability are among the key determinants of long-term sickness absence [3], intentions to retire early [4] and early exit from work [5], and is associated with health outcomes both during working life and following retirement [6,7,8,9].

Work ability has multicausal determinants. The personal characteristics of workers, family and social factors, working conditions and work organization have been identified as associated with work ability [10]. Previous research has shown that work ability is negatively influenced by high physical work demands, high psychosocial work demands (e.g., lack of possibilities to control one’s own work) and unhealthy lifestyles (e.g., lack of physical activity) [11,12,13,14,15]. 

Maintaining the ability to work of all employees by identifying and targeting modifiable factors has become an important topic in research [16,17], although it could be argued that some groups warrant greater attention than others. Specifically, there is evidence to suggest that gender, age and the nature of one’s profession may impact upon work ability [18,19]. 

With regards to gender, the evidence identifies females as an important group of interest. With just some exceptions [20], the majority of studies have shown that women have worse work ability indices [11,12,13,14,15,21] and that factors affecting work ability are different in males and females [22]. Various models have been proposed to explain gender inequalities in numerous indicators of health and quality of life. Such models include the hypothesis of differential vulnerability and exposure [23]. The exposure hypothesis states that because women and men differ with respect to access to material resources and other social conditions of life, they are exposed to different levels of risk, which in turn results in different health outcomes [23,24]. The vulnerability hypothesis asserts that women and men react differently to various conditions of life and thus differ in their vulnerability to risk factors [25]. 

In relation to age, research across countries and occupations has generally found work ability to be negatively associated with age [26]. This association has mainly been found in occupational sectors with highly physically demanding work [11]. On the other hand, although some psychosocial factors, such as job control, reward and social support, exert a protective effect over work ability [27,28], recent studies have suggested that the effect of these factors on well-being may vary according to workers’ ages [29]. Given that we are witnessing a progressive aging of the working population [30], it is important to deepen knowledge of the factors that mediate the relationship between age and work ability. 

A final group to be considered concerns those sectors or professional groups that typically present diminished work ability. Health center employees, in both primary and hospital care, but particularly hospital workers, are exposed to particularly demanding situations that could detract from their work ability [31,32]. Given the importance of these employees, particularly in light of the current health crisis due to the COVID-19 pandemic, they represent another sub-group of interest. 

Finally, most studies suggest that work ability worsens with increasing age, whilst also being worse in women. However, other investigations [20] disagree in this regard, suggesting that gender and age differences only appear under certain conditions; in other words, they are dependent upon other mediating or moderating factors [11]. The main contribution of the present paper, therefore, is intended to offer a different perspective by exploring which specific “third factors” mediate the associations between gender, age, professional group and work ability. 

In consideration of this, the present study has two main objectives. The first is to study the differences in work ability in health center workers in relation to the determinants of age, sex and professional group. A further objective is to examine whether factors related to working conditions mediate these aforementioned associations.

## 2. Materials and Methods

Workers from six public health centers attending the occupational health department of participating health centers (three hospitals, two primary health care centers and a combined health facility) to complete a routine health check were contacted. A member of the research team working at each center explained the aims, procedures and ethical considerations of the study. Participants were included in the study on a voluntary basis after signing an informed consent form. In total, 1184 workers (70% of workers invited to participate) answered the questionnaire. Data collection was conducted throughout 2019. 

Workers were excluded from the study if they had any other reason for attending the center other than a routine health check-up. 

All subjects gave their informed consent for inclusion before they participated in the study. The study was conducted in accordance with the Declaration of Helsinki, and the protocol was approved by the Andalusian research ethics committee (REC of the Virgen Macarena-Virgen del Rocio university hospitals: d9b449426c41062448a2d8be713a0b063741ae96).

The following variables and instruments were used: 

Dependent variables: The work ability index (WAI) provided the dependent variable. The term work ability was first introduced in Finland in the 1980s; the Finnish Institute of Occupational Health conceived an instrument to assess and monitor work ability [33] in groups of interest. The Spanish version of the WAI was used following translation of the English version [33]. The WAI is composed of seven indicators. Indicators WAI1, WAI3, WAI4, WAI5 and WAI6 comprise just one item, item WAI2 comprises two items and item WAI7 is described by three items. Different response formats and different scales are used for each indicator. Outcomes for the seven indicators are therefore transformed before being summed to produce an overall score. Final scores range between 7 and 49, with higher values indicating better work ability. Respondents were grouped according to their scores: poor (7–27), moderate (28–36), good (37–43) and excellent work ability (44–49). 

Independent variables: Gender, age and the professional category to which participants belonged (dichotomized as “health worker” or “not a health worker”). Health workers pertained to professionals from the following groups of staff: physicians, nursing, nursing assistant and nursing technicians. Non-health staff included administrative, caretaker, maintenance and general services staff (for example those in kitchen or laundry services).

Mediators: Occupational risk variables were considered as potential meditators of the relationships between WAI and the independent variables (age, gender and professional group). The following types of risk were analyzed: (a) environmental risk; (b) ergonomic risk; (c) psychosocial risk; (d) violence and discrimination; and (e) work–family conflict [34]. 

Environmental and ergonomic risk each consisted of 4 items rated along a 4-point scale from “never” to “always”. Ergonomic risk relates to the aspects of a job or task that impose a biomechanical stress on the work. They are likely to cause or contribute to awkward or static postures, cold temperatures, contact stress, force, repetition and vibration, posing injury risks to the musculoskeletal system. Environmental risk relates to challenges posed by one’s surroundings at work. The stem “In the tasks that you perform in your work, to what extent are you exposed to…?” preceded all items. Respondents were prompted to respond to each item twice, once in relation to their current condition and, next, in relation to the length of their working career. An example of environmental risk included “extreme environmental conditions”, whilst an example of ergonomic risk was “I work in static, painful or tiring positions”. Items for each were summed to provide possible scores of between 4 and 16. Higher scores indicated greater risk. Psychosocial risk relates to negative psychological responses to work and workplace conditions and incorporates relationships with colleagues and supervisors. Psychosocial risk composed 14 items, of which 11 began with the stem “In your work, how frequently…?” followed by examples of risk situations, e.g., “have to work very fast”. Responses were given on a 5-point scale ranging from “never” to “always”. The final three items targeted aspects related with the way in which they are managed, e.g., “To what extent do you think you are fairly treated at your workplace?”. These items are rated on a 5-point scale from “to a large extent” to “to no extent”. Framing varied, thus, where required items were transformed so that higher scores indicated greater risk. Score were summed producing final values of between 14 and 35. The discrimination/violence dimension consisted of seven items with two targeting violence (e.g., “To what extent do you think you have suffered situations of psychological violence (threats, insults etc.)?”) and five targeting discrimination (e.g., “to what extent have you suffered sexual harassment?”). Items were reported on a 5-point scale ranging from “never” to “regularly”. Scores were summed with higher scores indicating greater risk. Scores ranged from 7 to 35. Three items were used to estimate work–life conflict. Items were rated on a 5-point scale from “always” to “never” and were summed, with higher scores showing worse work–family conflict. 

For statistical analyses, Pearson point-biserial correlations were performed between overall WAI and the four binary sociodemographic variables (gender, age and professional category) to establish the presence of independent relationships (see Appendix A). Following this, a mediation analysis was conducted using the approach described by Sobel (1982) [35], with occupational risk variables and work–family conflict being the potential mediators. This approach follows three stages. Firstly, unstandardized regression coefficients and their associated error were calculated for the relationship between the independent variable (gender, age and professional category) and the mediator. As all mediators were continuous, linear regression was used for this stage. In the second stage, the total effect (and error) of the mediator on the dependent variable was calculated, whilst controlling for the independent variable. Logistic regression was used at this stage as the dependent variable (WAI) was categorical. WAI was dichotomized to ease interpretation, with poor and moderate scores, and good and very good scores being grouped, respectively. Homoscedasticity was not a problem. In mediation, the relationship between the independent variable and the dependent variable is hypothesized to be an indirect effect that exists due to the influence of a third variable (the mediator). Thus, in the final stage, this indirect effect was calculated from the coefficients calculated in Stages 1 and 2 using the Sobel test. The Sobel test reveals the magnitude of mediation due to the mediator. Following Kenny et al. (1998) [36], it is not necessary for the independent variable to independently predict the outcome for mediation to occur. Thus, the Sobel test was conducted between all study variables and not only when independent relationships were established. Crude coefficients were used following the findings reported by [37] in relation to logistic regression-based mediation analysis with dichotomous outcomes and continuous mediators. 

The statistical power of each test was calculated using the joint test of significance described by Hayes and Scharkow (2013) [38]. The statistical program SPSS version 26.0 was used to conduct the mediation analyses, whilst MedPower [39] was used to calculate statistical power.

## 3. Results

The power for each test ranged between 66.5% and 100%, with most tests having more than 80% power, suggesting that the study was sufficiently powered for the analysis. The questionnaire responses according to gender and professional category are presented in Table 1.

Both gender and age were significantly correlated with WAI. Lower WAIs were seen in females (r^pb^ = −0.148; *p* < 0.001) and with increasing age (r^pb^ = −0.217; *p* < 0.001). Professional group (r^pb^ = −0.027; *p* = 0.459) was not independently associated with WAI.

Stage one of the mediation analysis revealed that females and health care workers reported higher indices (greater risk) for all included risk variables (mediators), with outcomes being significant in most cases (*p* < 0.05). In the case of age, the risk of suffering violence and discrimination significantly increased with age (*p* < 0.05). Stage two was proceeded to when stage one outcomes were significant (see Table 2, Table 3 and Table 4). When adjusting for gender, greater current ergonomic, historic ergonomic, psychosocial and work–family conflict all predicted worse WAI (*p* < 0.001). Females were less likely to be in the better WAI group even when current and historic ergonomic risk, and psychosocial risk were included in the model, but not when work–family conflict was. When adjusting for age, only violence and discrimination predicted WAI (*p* < 0.05). Finally, when adjusting for professional category, all potential mediators apart from current ergonomic risk predicted WAI (*p* < 0.01). 

Analysis using the Sobel test produced a number of significant results (all results presented in Table 2, Table 3 and Table 4). The relationship between gender and WAI was mediated by current ergonomic risk (*t* = 2.80; SE = 0.01; *p* < 0.01), historic ergonomic risk (*t* = 3.19; SE = 0.06; *p* < 0.01), psychosocial risk (*t* = 3.80; SE = 0.05; *p* < 0.001) and work–family conflict (*t* = 5.14; SE = 0.07; *p* < 0.001). 

The relationship between age and WAI was mediated by violence/discrimination (*t* = −2.17; SE = 0.00; *p* < 0.05) comparisons. 

Mediated relationships were seen between professional group and WAI when current environmental risk (*t* = −2.51; SE = 0.03; *p* < 0.05), historic environmental risk (*t* = −3.45; SE = 0.06; *p* < 0.01), historic ergonomic risk (*t* = −2.93; SE = 0.07; *p* < 0.01), psychosocial risk (*t* = −2.81; SE = 0.05; *p* < 0.01), violence-discrimination (*t* = −2.98; SE = 0.03; *p* < 0.001) and work–family conflict (*t* = −4.55; SE = 0.07; *p* < 0.001) were included in the model. 

## 4. Discussion

Firstly, our results corroborate other studies that have shown women and older individuals to have worse work ability [15,21]. These findings are unsurprising given that females have been shown to have a lower WAI than males in other working contexts [40,41]. However, the present study also found variables to exist that mediated this association. Indeed, the worse WAI in females relative to males was mediated by the fact that women were exposed to a greater extent to current and historic ergonomic risk and work–family conflict. Of these variables, the latter emerged as the strongest mediator as gender-based differences in work ability disappeared when work–family conflict was controlled. In this sense, other studies have shown the importance of especially work–family conflict to better understand the impinged work ability amongst women [22,42].

Gragnano et al. (2020) [43] also recently identified that females rate the family aspect of work–family balance higher than other aspects, with work–family balance influencing job satisfaction through the WAI. Gender inequalities in work–family conflict have been conceptualized as a key element for understanding potential gender inequalities in relation to different health outcomes [44]. In this respect, two alternative hypotheses have been proposed. The first is based on gender differences in vulnerability to family and work stressors (including work–family conflict) [45] and the second concerns differential gender exposure to situations of imbalance between life spheres [46,47].

Whilst evidence of gender differentiated vulnerability is difficult to gather, there is some evidence to support the hypothesis of differential exposure. The role of women in contemporary society is diverse. In Spain, whilst men engage 5 h more a week in paid work than females, women spend on average 13 h more a week than men engaged in all paid and unpaid work [48]. In addition to these concerning national statistics, there is broader evidence that work integration policies are not working for women. Commitment of the European community [49] to equality of opportunities in various societal spheres has seen an increase in the introduction of work–life and family-friendly work policies. With females traditionally being the primary caregiver within families, such policies are designed to ease their path into work. However, there is some evidence that these approaches, including flexible/alternative working arrangements, paid and unpaid leave arrangements, dependent care services and access to information, resources or services, are falling short. For instance, Warren (2004) [50] suggested that, despite efforts, female part-time employees are more likely to be financially insecure, less satisfied with their work–family balance and more likely to intend to quit their job than full-time colleagues. Brough et al. (2008) [51] suggests that workplaces are reluctant to reorganize and enable such policies to be implemented. Thus, work–family balance policies seem better equipped at promoting the “life” side (typically interpreted as “family life” with males being mainly targeted due to their lesser engagement in child rearing, etc.) than the “work” side. As a result, the door to the workplace may remain firmly shut for some women, whilst those females who are gainfully employed may still be at greater risk of suffering from adverse psychological outcomes.

With regards to the associations between work ability and age, older individuals in the present study had a lower WAI than younger colleagues. This relationship remained significant even when violence or discrimination was considered, though violence–discrimination emerged has strengthening this relationship. Other studies have found similar associations regarding work ability, for instance between ageism and older workers’ retirement plans [52].

Despite the emergence of laws to prevent age discrimination in the workplace, such discrimination is still frequent [30,53]. Our findings indicate that this dynamic has had a particularly negative impact on the work ability of the older adults working in health centers in the present study. 

When we consider professional category (health workers vs. non-health workers), no significant difference in work ability was found. However, when violence/discrimination and work–family conflict are considered as mediators, health workers are more likely to belong to the poorer WAI group. Health care workers are particularly exposed to violence or discrimination [54], with evidence of negative implications of this on psychosocial demand and organizational justice [54]. Violence has also been shown to impair work ability [55,56], with violence not having to be physical to have negative work-related implications [57]. Exposure to violence or discrimination likely impacts upon victims’ perceptions of other work-related factors. In the present study, experiences of violence and discrimination meant that health workers had a lower WAI than their counterparts not working in these settings. 

Within health workers, work–family conflict was the strongest mediator to emerge of WAI. Previous studies have identified some implications of working in the health setting [58]. These include a lack of flexibility in the working day, managing patient expectations and difficulties in taking time off work, particularly at short notice. Importantly, these aspects of striking a positive balance between the time spent working and engaged in other non-work activities appear to be more deficient in females and health workers, with consequent negative implications for the work ability of these groups. 

A number of limitations of the present research should be indicated. The study is cross-sectional in nature and so causal conclusions cannot be made. However, it reports the first-wave cohort data with the second wave of data collection being planned for later in the year. It therefore provides important first insights with the opportunity for later verification. A further limitation is that the statistical procedure applied lacks power relative to other approaches to mediation analysis. For this reason, further outcomes of interest may have been missed; however, the analysis procedure was selected based on its appropriateness to the collected data. Finally, examination of the interceding influence of “third factors” as potential mediators was exploratory in nature. The statistical approach to determine whether a variable is a mediator or moderator is limited and such decisions are better made when framed by prior theoretical or empirical evidence. Despite these limitations, the study benefits from highly novel data collected within a relatively large sample. Whilst the inclusion of only health care workers limits the generalizability of the findings to the wider population, it is important to examine high-risk employees who play a critical role in public health, particularly in the present time with a health pandemic. A future perspective of the present study would be to conduct follow-up analysis within specific groups of health staff (e.g., medical versus nursing staff) in order to identify differences. 

## 5. Conclusions

The present study identified the importance of risk variables for the work ability of health workers according to gender, age and professional group. Females older adults and health workers had a lower work ability after accounting for various aspects of risk. Perception of work–life balance and violence–discrimination may be particularly important and should be considered by interventions targeting improvements in work ability. 

## Figures and Tables

**Table 1 ijerph-18-00877-t001:** Study sample characteristics: overall and by gender and professional category.

Variables	Overall	Gender		Professional Category	
		Female	Male	Health Workers	Non-Health Workers
*n* (%)	1184 (100)	796 (67.2)	324 (27.4)	757 (63.9)	332 (28.0)
*mean* (*SD*)					
Age	54.46 (5.64)	54.13 (5.52)	55.11 (5.89)	54.44 (5.81)	54.83 (5.23)
WAI score	37.77 (7.20)	37.00 (7.20)	39.30 (6.95)	37.57 (7.14)	38.00 (7.36)
Current environmental risk	5.94 (1.94)	5.98 (1.96)	5.93 (1.93)	6.17 (2.06)	5.58 (1.72)
Historic environmental risk	6.68 (2.17)	6.76 (2.19)	6.58 (2.11)	7.05 (2.25)	5.83 (1.74)
Current ergonomic risk	9.19 (2.79)	9.36 (2.84)	8.76 (2.61)	9.29 (2.81)	8.92 (2.76)
Historic ergonomic risk	9.59 (2.70)	9.87 (2.74)	9.07 (2.54)	9.82 (2.76)	9.08 (2.51)
Psychological risk	42.36 (6.95)	42.96 (6.93)	40.88 (6.93)	42.92 (6.63)	41.47 (7.69)
Violence and discrimination	9.55 (2.71)	9.62 (2.80)	9.53 (2.57)	9.83 (2.89)	9.16 (2.31)
Work–family conflict	7.60 (2.33)	7.92 (2.36)	6.97 (2.23)	7.91 (2.37)	7.11 (2.20)

**Table 2 ijerph-18-00877-t002:** Mediation analysis results of the relationship between gender (females relative to males) and the work ability index (WAI).

Mediator	Stage 1				Stage 2				Sobel Test		
	β1	B	SE	*p*	β2	B2	SE	*p*	*t*	SE	*p*
Current environmental risk	0.048	0.011	0.137	0.725							
Historic environmental risk	0.176	0.038	0.177	0.319							
Current ergonomic risk	0.602	0.098	0.193	0.002	−0.211	0.810	0.033	0.000	2.803	0.045	0.005
Historic ergonomic risk	0.806	0.139	0.218	0.000	−0.247	0.781	0.039	0.000	3.193	0.062	0.001
Psychosocial risk	2.078	0.136	0.472	0.000	−0.098	0.907	0.013	0.000	3.802	0.054	0.000
Violence-discrimination	0.091	0.015	0.181	0.615							
Work–family conflict	0.948	0.183	0.154	0.000	−0.391	0.676	0.042	0.000	5.135	0.072	0.000

**Table 3 ijerph-18-00877-t003:** Mediation analysis results of the relationship between age and the WAI.

Mediator	Stage 1				Stage 2				Sobel Test		
	β1	B	SE	*p*	β2	B2	SE	*p*	*t*	SE	*p*
Current environmental risk	−0.014	−0.041	0.011	0.197							
Historic environmental risk	−0.018	−0.045	0.015	0.236							
Current ergonomic risk	0.003	0.007	0.016	0.835							
Historic ergonomic risk	−0.006	−0.012	0.019	0.757							
Psychosocial risk	0.075	0.061	0.039	0.054							
Violence-discrimination	0.036	0.074	0.015	0.015	−0.142	0.868	0.028	0.000	−2.169	0.002	0.030
Work–family conflict	0.008	0.019	0.013	0.530							

Note: Stage 1 = path (a) from the IV to the mediator. Stage 2 = path (ab) from the IV to the DV adjusted for the mediator. β1 and B: Raw (unstandardized) regression coefficients and standardized regression coefficients, respectively, between the independent variable (gender, age, professional group) and the mediator. β2 and B2: Raw (unstandardized) regression coefficients and standardized regression coefficients, respectively, between the mediator and WAI controlling for the independent variable (gender, age, professional group). Note: Stage 1 uses linear regression and so B > 0 is positive and B < 0 is negative. Stage 2 uses logistic regression and so B2 > 1 is positive and B2 < 1 is negative.

**Table 4 ijerph-18-00877-t004:** Mediation analysis results of the relationship between professional group (health care workers relative to non-health workers) and the WAI.

Mediator	Stage 1				Stage 2				Sobel Test		
	β1	B	SE	*p*	β2	B2	SE	*p*	t	SE	*p*
Current environmental risk	0.584	0.136	0.137	0.000	−0.124	0.883	0.040	0.002	−2.507	0.029	0.012
Historic environmental risk	1.214	0.251	0.179	0.000	−0.168	0.846	0.042	0.000	−3.445	0.059	0.001
Current ergonomic risk	0.365	0.060	0.195	0.061							
Historic ergonomic risk	0.747	0.124	0.228	0.001	−0.257	0.773	0.039	0.000	−2.934	0.065	0.003
Psychosocial risk	1.451	0.096	0.479	0.003	−0.098	0.906	0.013	0.000	−2.812	0.051	0.005
Violence-discrimination	0.665	0.112	0.181	0.000	−0.148	0.862	0.029	0.000	−2.982	0.033	0.003
Work–family conflict	0.792	0.156	0.154	0.000	−0.421	0.657	0.043	0.000	−4.553	0.073	0.000

Note: β1 and B: Raw (unstandardized) regression coefficients and standardized regression coefficients, respectively, between the independent variable (gender, age, professional group) and the mediator. β2 and B2: Raw (unstandardized) regression coefficients and standardized regression coefficients, respectively, between the mediator and WAI controlling for the independent variable (gender, age, professional group). Note: Stage 1 uses linear regression and so B > 0 is positive and B < 0 is negative. Stage 2 uses logistic regression and so B2 > 1 is positive and B2 < 1 is negative.

## Data Availability

The data presented in this study are available on request from the corresponding author. The data are not publicly available due to privacy reasons specified in the informed consent signed by the participants and participating institutions.

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
