# Peer review of "Mediational Occupational Risk Factors Pertaining to Work Ability According to Age, Gender and Professional Job Type"

_ijerph, 2021, doi:10.3390/ijerph18030877_

Round 1
Reviewer 1 Report
Comment on ijerph-1019819-peer-review-v1
About your two research objectives
- Since the relationships between gender, age and occupational job type and work ability are well documented in the literature, you should explicate why we need another study on their relationship. How does your study add new value to the literature? Please articulate the research problems and your contributions clearly.
- You examine whether factors related to working conditions mediate or explain the relationships. Working conditions serve as the moderators instead of mediators in your research model. You mix moderators with mediators. Working conditions are boundary conditions that influence the relationship between employee demo and work ability. Mediators are intermediate variables that links the relationship between independent variables and outcome variables.
- Offer the hypotheses at the end of Introduction part.
- If you aim to find the link between employee demo and work ability, the mediator should act as a bridge that explains why employee demo influences work ability. For example,
Hypothesis 1: Employee skill mastery mediates the relationship between employee age (or tenure) and work ability.
- Compared with working conditions as the moderators, it is better for employee demo to act as the moderators in the relationship between working conditions and work ability. For example,
Hypothesis 2: Environmental risk attenuates employee work ability.
Hypothesis 3: Employee gender moderates the relationship between environmental risk and work ability such that the negative relationship is stronger for women than for men.
Hypothesis 4: Employee tenure moderates the relationship between environmental risk and work ability such that the negative relationship is stronger for employees with short tenure than for those with long tenure.
Hypothesis 5: Employee job type moderates the relationship between environmental risk and work ability such that the negative relationship is stronger for health worker than for non-health worker.
- For Results, please provide mean, standard deviation 9SD) and correlations of study variables.
- To test mediation, you can use PROCESS. For PROCESS, please google Prof. Hayes, mediation.
- There is no need to change DV into categorical variable.
- PROCESS can test one IV, multiple mediators and one DV, simultaneously. You can get path coefficients from IV to mediators (a), and from mediators to DV (b). You can also get the indirect effect from IV to DV via mediators (ab).
- PROCESS can also test moderating effect. IV and moderators can be categorical variables.
- Your test of mediation is wrong. For simple mediation test, make sure the significant path from IV to Mediator (a), and the significant path from Mediator to DV (b) (You have already done!), but more important, the significant indirect effect of IV on DV via mediators (ab). PROCESS can provide you with these results. However, your mediation hypotheses are not correct, as noted in point 2.
- Add the headers of theoretical and practical implications in the Discussion. Use “First, ---. Second, ---. Third, ---.” for clarity.
Author Response
Comments from reviewer 1
- Comment: Since the relationships between gender, age and occupational job type and work ability are well documented in the literature, you should explicate why we need another study on their relationship. How does your study add new value to the literature? Please articulate the research problems and your contributions clearly.
Response: We have provided further clarification on this point which can be found on lines 75-80.
- Comment: You examine whether factors related to working conditions mediate or explain the relationships. Working conditions serve as the moderators instead of mediators in your research model. You mix moderators with mediators. Working conditions are boundary conditions that influence the relationship between employee demo and work ability. Mediators are intermediate variables that links the relationship between independent variables and outcome variables.
Response: The Sobel test of mediation was used in the present study as a non-parametric handling of data was required. In this model, variables are not pre-determined as mediators or moderators. Instead, the behaviour of coefficients within the model determines whether each variable acted in a mediating or moderating way.
- Comment: Offer the hypotheses at the end of Introduction part.
- If you aim to find the link between employee demo and work ability, the mediator should act as a bridge that explains why employee demo influences work ability. For example,
Hypothesis 1: Employee skill mastery mediates the relationship between employee age (or tenure) and work ability.
- Compared with working conditions as the moderators, it is better for employee demo to act as the moderators in the relationship between working conditions and work ability. For example,
Hypothesis 2: Environmental risk attenuates employee work ability.
Hypothesis 3: Employee gender moderates the relationship between environmental risk and work ability such that the negative relationship is stronger for women than for men.
Hypothesis 4: Employee tenure moderates the relationship between environmental risk and work ability such that the negative relationship is stronger for employees with short tenure than for those with long tenure.
Hypothesis 5: Employee job type moderates the relationship between environmental risk and work ability such that the negative relationship is stronger for health worker than for non-health worker.
Response: We acknowledge the suggestion to add hypotheses to the introduction and have done so on page 2, lines 85-89. We appreciate the time taken by the reviewer to outline some interesting potential hypotheses. We have taken its suggestions to structure 3 hypotheses around the 3 main observed sociodemographic variables (age, gender and professional category). As the study is exploratory in nature, no hypothetical assumptions were made regarding the type of influence (mediation or moderation) and the model was not configured towards either moderation or mediation. Instead, open hypotheses were made and the presence of mediation or moderation was freely interpreted from statistical outcomes.
- Comment: For Results, please provide mean, standard deviation 9SD) and correlations of study variables.
Response: All of this information can be found in Tables 1 (means and SD given for all outcomes) and 2a and 2b (standardised and non-standardised coefficients given for all study variables).
- Comment: To test mediation, you can use PROCESS. For PROCESS, please google Prof. Hayes, mediation.
Response: In order to use PROCESS, WAI would have to be treated as a continuous variable and data would have to meet a number of assumptions. Data did not meet assumptions of 1) normality (the residuals associated with WAI (when handled as a continuous variable) and the IVs did not show a normal association); 2) multivariate normality (multivariate asymmetry and kurtosis were not consistent with a normal multivariate distribution). For this reason, parametric mediation analysis was not appropriate and, as is consistent with the handling of this variable in the literature, WAI was treated as a categorial variable to conduct relevant non-parametric analysis.
- Comment: There is no need to change DV into categorical variable.
Response: As discussed above, the DV had to be handled categorically.
- Comment: PROCESS can test one IV, multiple mediators and one DV, simultaneously. You can get path coefficients from IV to mediators (a), and from mediators to DV (b). You can also get the indirect effect from IV to DV via mediators (ab). PROCESS can also test moderating effect. IV and moderators can be categorical variables.
Response: Whilst we acknowledge this to be the case, our data was not appropriate for this type of analysis.
- Comment: Your test of mediation is wrong. For simple mediation test, make sure the significant path from IV to Mediator (a), and the significant path from Mediator to DV (b) (You have already done!), but more important, the significant indirect effect of IV on DV via mediators (ab). PROCESS can provide you with these results. However, your mediation hypotheses are not correct, as noted in point 2.
Response: Tables 2a and 2b present the mediation analysis. The steps described above were all completed (information can be found on lines 158-162 and 165-168), however, as step b is not necessary for mediation to exist (unless only complete mediation is expected [Kenny et al, 1998]), only steps (a) and (ab) are presented in the tables (stage 1 and stage 2). We have added a footnote to the tables to clarify this information.
- Comment: Add the headers of theoretical and practical implications in the Discussion. Use “First, ---. Second, ---. Third, ---.” for clarity.
Response: The implications of the work are now discussed around the three stated hypotheses (line 215, 252 and 270)

Reviewer 2 Report
This paper is of interest mainly because of the results that work-life balance and discrimination are important mediators for work ability. These association has been studied scarcely and need further attention.
The manuscript need, however, some clarifications especially according to the methods chapter.
The sample is divided in health and non- health worker. On what basis and what occupational groups are included (at least examples) and especially who are in the non-health worker group. Why are non- health workers included as a group. Where all in both groups women and do they come from hospitals all of them. Health center means in many countries something else than a hospital. In the introduction mainly hospital workers are mentioned. Do you mean nurses or other? The work ability among nurses and assistant nurses may differ for example? Age of the sample is needed also in the abstract.
There are several variables used as determinants for work ability but including work- life balance and violence/discrimination under work conditions/ occupational risks warrants an explanation.
In the methods it is important to name the variables and use them as such. No explanation is now on what is e.g. historic environmental or ergonomics etc. Tables for the variables would be more easy to read.
There are some limitations which are discussed but how about strengths?
You write both work ability and workability. I prefer the first way.
More commnets in the attached file.

Author Response
Response to comments from Reviewer 2
- Comment: This paper is of interest mainly because of the results that work-life balance and discrimination are important mediators for work ability. These association has been studied scarcely and need further attention.
Response: Thank you for this comment.
- Comment: The manuscript need, however, some clarifications especially according to the methods chapter. The sample is divided in health and non- health worker. On what basis and what occupational groups are included (at least examples) and especially who are in the non-health worker group. Why are non- health workers included as a group. Where all in both groups women and do they come from hospitals all of them. Where all in both groups women and do they come from hospitals all of them. Health center means in many countries something else than a hospital. In the introduction mainly hospital workers are mentioned. Do you mean nurses or other?. The work ability among nurses and assistant nurses may differ for example?
In relation to comment: The manuscript need, however, some clarifications especially according to the methods chapter.
Response: Further clarifications have been provided within the methods section on lines 119-122, 129-132, 138-140. Hypotheses have also been added to the end of the introduction (lines 85-89) which should aid understanding of the methods.
In relation to comment: The sample is divided in health and non- health worker. On what basis and what occupational groups are included (at least examples) and especially who are in the non-health worker group.
Response: This information has now been added on lines 119-122.
In relation to comment: Why are non- health workers included as a group.
Response: Existing literature on WAI suggests that “Health employees, particularly hospital workers, are exposed to particularly demanding situations which could detract from their work ability [31,32]. Thus, it was of interest to us to examine whether individuals working in health centres presented different WAI as a function of whether they held a health-related job position or a non-health related job position.
In relation to comment: Where all in both groups women and do they come from hospitals all of them.
Response: There was an even distribution of males and females across both professional groups, with participants coming from a mix of hospitals and health centres. We have sought to clarify this on line 71-72.
In relation to comment: Health center means in many countries something else than a hospital. In the introduction mainly hospital workers are mentioned. Do you mean nurses or other?
Response: We have modified the sentence on line 71-72 to clarify this issue. The sentence now reads “Health centre employees, in both primary and hospital care but particularly hospital workers, are exposed to particularly demanding situations which could detract from their work ability [31,32]”.
In relation to comment: The work ability among nurses and assistant nurses may differ for example?
Response: We acknowledge this comment and have added as a future perspective that it would be of interest to conduct a follow-up analysis in order to analyse potential differences between more specific groups, for example between medical staff and nursing staff, and between nursing staff and nursing assistant staff (lines 288-290).
- Comment: Age of the sample is needed also in the abstract.
Response: Age has now been added to the abstract (line 22).
- Comment: There are several variables used as determinants for work ability but including work- life balance and violence/discrimination under work conditions/ occupational risks warrants an explanation.
Response: work-life balance and violence/discrimination are widely acknowledged as being occupational psychosocial risk factors (e.g. Chirico, 2017: “broader “psychosocial risk,” which includes new and emerging psychosocial risk factors, such as the combined exposure to physical and psychosocial risks, job insecurity, work intensification and high demands at work, high emotional load related to burnout, work‐life balance problems, and violence and harassment at work”). This reference has now been added to line 127.
- Comment: In the methods it is important to name the variables and use them as such. No explanation is now on what is e.g. historic environmental or ergonomics etc. Tables for the variables would be more easy to read.
Response: We have modified this section as requested and provided greater detail on the mentioned variables (lines 129-132 and lines 138-140).
- Comment: There are some limitations which are discussed but how about strengths?
Response: We have now added a line regarding the strengths of the study to this paragraph (line 285-288).
- Comment: You write both work ability and workability. I prefer the first way.
Response: We have replaced workability with work ability throughout the manuscript.

Reviewer 3 Report
This is a well-designed and well-prepared research article. The authors successfully describe the mediation effect about many factors on the work ability. Overall, this manuscript is suitable for publication on the journal. But I still have some minor suggestion about the topics might highlight the study population is limited to healthcare workers. Because the healthcare workers are a special working population and the study conclusion cannot be totally generalized to other working population.
Author Response
Response to Reviewer 3 comments
- Comment: This is a well-designed and well-prepared research article. The authors successfully describe the mediation effect about many factors on the work ability. Overall, this manuscript is suitable for publication on the journal. But I still have some minor suggestion about the topics might highlight the study population is limited to healthcare workers. Because the healthcare workers are a special working population and the study conclusion cannot be totally generalized to other working population.
Response: We have now acknowledged this limitation in the limitations section (lines 285-290).
Round 2
Reviewer 1 Report
I look through the manuscript, but find that (1) there is no response letter that addresses each of my comments, (2) my comments are not addressed carefully. For example, I suggest using bootstrapping (through PROCESS tool) to calculate indirect effect, rather than Sobel test. (3) Hypotheses are wrong. I mentioned that the authors confuse mediators with moderators. In the revised version on page 2, authors use "influence (mediate or moderate)" without indicating it is a mediation or moderation relationship. I thus suggest (1) authors should provide a response letter that addresses each of my concerns; (2) authors should indicate whether you will test mediation or moderation; (3) authors should use bootstrapping rather than Sobel test mediation. (4) authors should follow the comments given in the prior review if they expect more positive feedback.Author Response
Response letter to reviewer 1
NOTE: Changes introduced in the manuscript corresponding to this revision are highlighted in “green”.
Comment: I look through the manuscript, but find that (1) there is no response letter that addresses each of my comments.
Response: A two-page response was elaborated which addressed ‘comment by comment’ the review made by reviewer 1 and submitted along with the revised version of the manuscript.
Comment: (2) my comments are not addressed carefully. For example, I suggest using bootstrapping (through PROCESS tool) to calculate indirect effect, rather than Sobel test.
Response: As explained in the prior response to this comment “In order to use PROCESS, WAI would have to be treated as a continuous variable and data would have to meet a number of assumptions. Data did not meet assumptions of 1) normality (the residuals associated with WAI (when handled as a continuous variable) and the IVs did not show a normal association); 2) multivariate normality (multivariate asymmetry and kurtosis were not consistent with a normal multivariate distribution). For this reason, parametric mediation analysis was not appropriate and, as is consistent with the handling of this variable in the literature, WAI was treated as a categorial variable to conduct relevant non-parametric analysis.”
Whilst we agree that bootstrapping via PROCESS is an excellent approach and provides rigorous outcomes (it would have also been much easier for us to have used this test over the Sobel approach) the test loses all rigour when applied under conditions in which its assumptions are violated (as would have been the case had we used it with our data) and cannot simply be used as a silver bullet.
Comment: (3) Hypotheses are wrong. I mentioned that the authors confuse mediators with moderators. In the revised version on page 2, authors use "influence (mediate or moderate)" without indicating it is a mediation or moderation relationship. I thus suggest (1) authors should provide a response letter that addresses each of my concerns;
Response: A full page of the response letter sent was dedicated to addressing reviewers’ concerns regarding mediation and moderation. We are very aware of the differences between moderation and mediation, with moderation referring to influences on the strength or direction of a relationship and mediation referring to a variable that explains the existence of a relationship between to other variables. Hypotheses were included at your prior request. They were not initially included because the analysis is exploratory in nature, hence, use of the term “influence”.
Comment: (2) authors should indicate whether you will test mediation or moderation.
Response: there is no need for hypotheses and "checking whether a variable is a mediator or a moderator" is not appropriate. We do not feel it necessary to indicate this as the study is exploratory in nature.
Comment: (3) authors should use bootstrapping rather than Sobel test mediation.
Response: As explained in our previous response letter in response to this comment, “In order to use PROCESS, WAI would have to be treated as a continuous variable and data would have to meet a number of assumptions. Data did not meet assumptions of 1) normality (the residuals associated with WAI (when handled as a continuous variable) and the IVs did not show a normal association); 2) multivariate normality (multivariate asymmetry and kurtosis were not consistent with a normal multivariate distribution). For this reason, parametric mediation analysis was not appropriate and, as is consistent with the handling of this variable in the literature, WAI was treated as a categorial variable to conduct relevant non-parametric analysis.” Unfortunately, this approach is not appropriate for our data.
Comment: (4) authors should follow the comments given in the prior review if they expect more positive feedback.
Response: We respect all comments received by all reviewers as a valuable means to improving manuscript quality. However, comments are suggestions and do not have to be followed blindly. We have fully considered all comments, deciding to include some and deciding to omit others, always providing full justifications as to why.
